# Tracking control of robotic manipulator end-effector trajectory based on robust sliding mode method

**Beibei Su**[ID]*

Wuxi Vocational College of Science and Technology, Jiangsu Wuxi, 214000, China

* moreover_edu@yeah.net

**Data availability statement:** All relevant data are within the manuscript and its Supporting Information files.

**Funding:** We acknowledge the financial support provided by the General Project of Philosophy and Social Sciences Research in Jiangsu Province Higher Education Institutions (Project Title: "Research on Entrepreneurial Models for College Students in Higher Vocational Colleges

## Abstract

This paper addresses the precise trajectory tracking of robotic manipulators (RMs) in automation tasks, particularly in hazardous environments. A dynamic model of the end-effector in Cartesian coordinates is developed to represent the system's motion within the workspace. A robust sliding mode control (SMC) law is proposed to enhance control performance, with a focus on ensuring stability and accuracy in trajectory tracking. The SMC law is designed by deriving the relationship between control inputs and joint torques. Numerical simulations demonstrate that the proposed control method achieves high accuracy in trajectory tracking, with significant reductions in tracking errors. The results quantitatively show improved performance in both path following and system robustness, making the method suitable for real-time industrial applications.

## 1 Introduction

A robotic manipulator (RM) is an advanced mechanical system designed to replicate the movements of the human hand [1–4]. Robotic manipulators are extensively utilized in various industrial applications, including automotive assembly and the manufacturing of electronic products. Their integration into production lines not only boosts efficiency and improves product quality but also minimizes the need for human labor. In particular, these robots are invaluable in environments that pose risks to human workers, as they can take over dangerous tasks, ensuring both safety and operational effectiveness throughout the production process [5–8].

Raza Ul Islam et al. [9] proposed an automated robotic system based on a serial manipulator, integrating image processing and inverse kinematics models to achieve precise object manipulation. The system is widely applicable to industrial tasks such as pick-and-place, sorting, and has the potential for expansion into large-scale industrial applications. Modeling a RM generally involves both dynamics and kinematics, utilizing mathematical formulations to represent the interactions between forces and motion. Additionally, 3D simulation tools such as MATLAB or SolidWorks are often employed to provide in-depth visualizations and analyze the system's behavior in greater detail [10–12]. In the area of trajectory tracking control

Based on 'Co-Creation between Teachers and Students'," Project No. 2024SJYB0726). The funders contributed to the data collection and analysis.

**Competing interests:** The authors have declared that no competing interests exist.

for RMs, scholars have proposed various control methods. Among the linear control methods, common ones include Proportional-Integral-Derivative (PID) control [13–15], adaptive control algorithms [16–18] or robust control [19–21], to ensure precision and stability in the manipulator's operation. These methods combined can optimize the design and performance of the RM, ensuring its efficiency and effectiveness in practical applications. For nonlinear control methods, researchers often utilize techniques such as sliding mode control (SMC) [22–24], feedback linearization [25], and dynamic surface control [26]. These approaches are effective in addressing the nonlinear dynamics of robotic manipulators, which enhances their ability to adapt to complex environments and improves the precision and stability of their trajectory tracking. Furthermore, there is a growing application of intelligent algorithms in RM control, such as genetic algorithms [27], fuzzy logic control [28], and neural network-based control [29]. By leveraging self-learning and adaptive capabilities, these algorithms optimize control strategies, significantly boosting the manipulator's trajectory tracking performance, particularly under dynamic and uncertain conditions, and offering superior robustness.

In [30], the application of PID controller in the field of robot tracking control driven by flexible actuators is discussed, and a simple and low-cost control strategy is developed by using the method of singular perturbation on three time scales. The experimental results show that this controller can not only significantly enhance the stability and anti-interference ability of the system, but also be especially suitable for those application environments with high spring stiffness. Khurram et al. proposed a fault-tolerant control (FTC) strategy that significantly enhances the performance and robustness of a five-degree-of-freedom (DoF) robotic manipulator in the presence of actuator and sensor faults. Their approach integrates a hybrid control scheme with redundant sensors and friction compensation, effectively mitigating the impact of faults on the system. By utilizing an adaptive backstepping methodology for fault estimation and employing a nominal control law for reconfiguration, this strategy ensures continuous high performance and system stability [31]. In addition, the control method proposed by Ullah et al. significantly enhances the trajectory tracking performance of the quadcopter throughout the entire flight process by incorporating functional link neural networks and a robust exact differentiator, especially performing excellently in the under-actuated case[32]. In [33], methods such as adaptive control and robust adaptive control are systematically introduced, covering their relationships and application scenarios, providing a concise guide to help students understand adaptive control in RMs. Similar to [33], Yuan et al. proposed an adaptive controller that addresses uncertainties in the dynamics of robots and motors using an adaptive nonlinear observer [34], achieving globally stable closed-loop control without acceleration feedback. Finally, the stability of the closed-loop system is further demonstrated using a Lyapunov function. In [35], Choubey et al. analyze the mathematical modeling and optimal path control of a three degree of freedom RM. The study involves observing three different continuous paths and implementing control actions using a PID controller based on Linear Quadratic Regulator (LQR). In [36], an advanced control algorithm combining Fast Integral Terminal Sliding Mode Control (FIT-SMC), Robust Exact Differentiator (RED) observer, and Feedforward Neural Network (FFNN) is proposed. This approach successfully enhances the performance of multi-degree-of-freedom robotic manipulators in friction compensation and trajectory tracking. Simulation results demonstrate the excellent convergence and robustness of the control system.

In the field of nonlinear control for RM trajectory tracking, Bagheri et al. designed a predictor-based controller for high degree of freedom robots to compensate for fixed input delays during pick-and-place tasks. Experimental and simulation results confirmed that the

controller effectively compensates for input delays, ensuring closed-loop stability of the system [37]. It is well known that SMC provides an effective control method for robotic manipulator trajectory tracking, especially suitable for applications in industrial automation that require high precision and robustness [38–41]. Safeer et al. proposed a neuro-fuzzy-based adaptive integral super-twisting sliding mode control method for the underactuated quadcopter. This approach effectively addresses the impact of uncertain disturbances and external forces. Specifically, the control law is designed to counteract model uncertainties and external disturbances, ensuring the stability of both linear and angular motions of the quadcopter during hover. By combining SMC with adaptive integral techniques, this method enhances the robustness of the system, allowing the quadcopter to dynamically adjust to the presence of unknown disturbances, thereby ensuring system stability! [42]. The adaptive backstepping-integral sliding mode control (AB-ISMC) method proposed by Afifa, Roohma [43], and others effectively improves the robustness and accuracy of speed control for the separately excited DC motor in the presence of parametric uncertainties and load disturbances, significantly reducing the settling time compared to traditional control methods. In [44], a method of non-singular terminal sliding mode tracking control for RMs is introduced, employing time delay estimation (TDE) without requiring any prior knowledge of the robot's dynamics, ensuring quick convergence and precise control. This approach's effectiveness, robustness, and efficiency have been demonstrated through simulations and practical experiments on both a 2-DOF planar robot and a 3-DOF PUMA-type robot. The robust backstepping integral sliding mode control (RBISMC) method proposed in Reference [45] significantly improves the stability and robustness of quadcopter control in terms of attitude, altitude, and position. Particularly in systems with high uncertainty, it demonstrates superior performance with faster convergence and chattering-free tracking. In [46], an innovative control strategy for robotic manipulator trajectory tracking, featuring a fractional integral SMC system based on Caputo-Fabrizio derivatives and atan Gana-Baleanu integrals. By employing a sliding surface that incorporates fractional calculus within the framework of an Euler-Lagrange derived mathematical model, this method significantly boosts the system's robustness and performance in the face of external disturbances and diverse operational tasks. The simulations indicate that this approach surpasses traditional integral sliding mode and optimal super-twisting SMC in managing external disturbances and trajectory shifts, offering superior control performance and enhanced energy efficiency. In [47], Safeer et al. proposed a fixed-time neuro-adaptive control method that combines non-singular fast terminal sliding mode control (TSMC) with an RBF neural network estimator. This approach successfully addresses the robustness issues in under-actuated nonlinear systems and has been validated through simulations for its effectiveness in real-world systems.

Intelligent algorithms are increasingly applied in RM trajectory tracking control, leveraging their ability to mimic human-like learning and adaptive optimization for precise motion control [6,11,29,48–50]. For instance, in [51], an adaptive neural network (NN) control strategy is proposed for RMs with multiple links and various constraints. This strategy effectively handles dynamics, state, input limitations, and unknown time-varying delays. NNs approximate nonlinearities, while time-varying barrier Lyapunov functions ensure system stability. Simulation results confirm the effectiveness of this approach. In [28], a task space control strategy is used to minimize the end-effector tracking error under dynamic uncertainties. Fuzzy logic networks simulate uncertain dynamics, and adaptive fuzzy components adjust the torque of a nonlinear proportional derivative control input, with experimental results demonstrating the strategy's effectiveness. In [52], a fuzzy logic controller is designed for a two-degree-of-freedom RM trajectory tracking task, using acceleration as the input linguistic

variable. Simulation experiments show that this fuzzy logic controller outperforms traditional PID controllers in trajectory tracking performance.

Based on our research and investigation into robotic manipulators (RMs), the main contributions of this study can be summarized as follows:

1. Development and validation of a dynamic model in Cartesian coordinates for the robotic arm's end-effector, providing valuable insights into spatial dynamics in real-world engineering applications.
2. Introduction of robust sliding mode control (SMC) strategies designed to enhance trajectory tracking accuracy and responsiveness, offering a significant improvement over conventional control methods.
3. Extensive numerical simulations that validate the effectiveness of the proposed SMC strategies in ensuring precise control of the end-effector's movement, thereby significantly enhancing operational accuracy.

The structure of the remaining sections of this paper is as follows: Section 2 describes the problem and the model transformation, providing the necessary lemmas. Section 3 first presents the control objectives of this study, then designs a robust sliding mode controller and conducts stability analysis of the controller. Section 4 uses MATLAB simulations to validate the effectiveness of the proposed control strategy. Finally, Section 5 summarizes the research and discusses future research directions.

## 2 Problem description and model transformation

### 2.1 Cartesian coordinates in workspace

**Remark 1:** The problem of converting the Cartesian coordinates $(x_1, x_2)$ of a robotic arm's end-effector in the workspace into the joint angles $(q_1, q_2)$ is known as the inverse kinematics of robots. As shown in Fig. 1, this is a schematic of a two degree of freedom RM. By determining the joint angles $q_1$ and $q_2$ from the position of the end-effector in the workspace, Document [53] provides detailed transformation methods and mathematical expressions. Solving this problem is crucial for achieving precise robot operations and path planning, involving complex geometric and algebraic calculations to ensure the arm accurately reaches predetermined positions.

**Assumption 1:** The robot's joints are ideal, without the influence of friction or other nonlinear effects. The robot's dynamic model is known, including the inertia matrix, Coriolis forces, and gravitational terms, and it is accurate within the entire operating range.

**Assumption 2:** External disturbances (such as load variations) are known and can be compensated for by a control method, or are considered as known disturbances in the error analysis. The controller design is based on minimizing steady-state errors, and it is assumed that the system will converge to the desired trajectory within a finite time.

**Assumption 3:** The robot operates within a specific workspace, and the position information of the end-effector can be accurately obtained through inverse kinematics transformation.

Based on Fig. 1, the specific position of the robotic arm's end-effector in the workspace can be determined

$$x_1 = l_1 \cos q_1 + l_2 \cos (q_1 + q_2) \tag{1}$$

$$x_2 = l_1 \sin q_1 + l_2 \sin (q_1 + q_2) \tag{2}$$

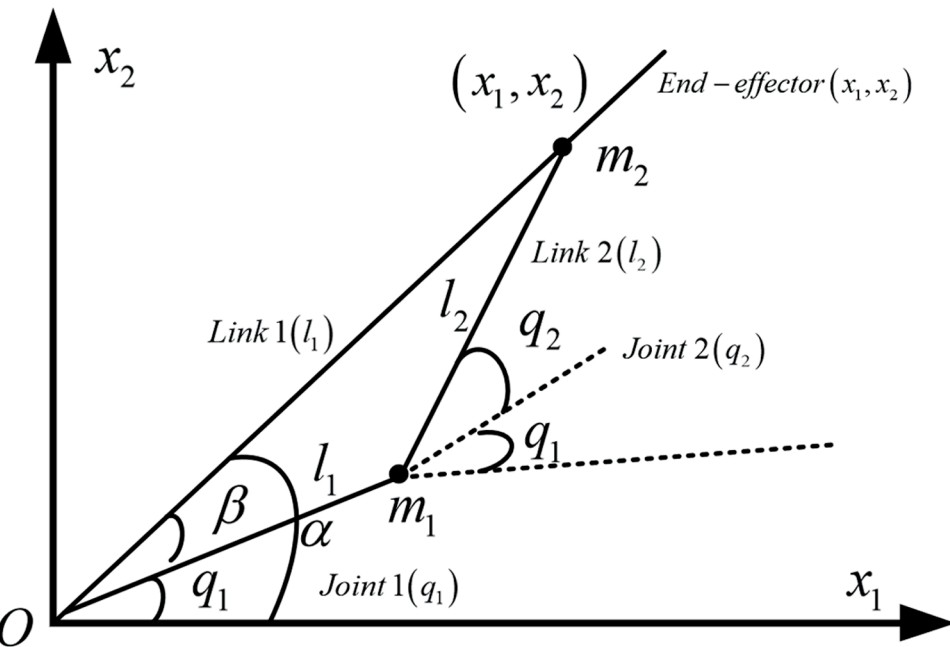

**Fig 1. The schematic diagram of two-degree-of-freedom robot manipulator.**

where $l_1$ and $l_2$ represent the lengths of the first and second links, while $q_1$ and $q_2$ are the angles of the first and second joints. The end-effector, marked as $(x_1, x_2)$, represents the working part of the RM. The diagram also labels the two joints (Joint 1 and Joint 2) and links (Link 1 and Link 2), along with the coordinate axes $x_1$ and $x_2$, representing the robot's workspace.

The square sum of Eqs. (1) and (2) can be obtained

$$x_1^2 = l_1^2 \cos^2 q_1 + 2l_1 l_2 \cos q_1 \cos (q_1 + q_2) + l_2^2 \cos^2 (q_1 + q_2) \tag{3}$$

$$x_2^2 = l_1^2 \sin^2 q_1 + 2l_1 l_2 \sin q_1 \sin (q_1 + q_2) + l_2^2 \sin^2 (q_1 + q_2) \tag{4}$$

By adding Eqs. (3) and (4) together and simplifying using the trigonometric identity $\cos^2 \theta + \sin^2 \theta = 1$, we can further derive

$$\begin{aligned} x_1^2 + x_2^2 &= l_1^2 \left( \cos^2 q_1 + \sin^2 q_1 \right) + 2l_1 l_2 \left( \cos q_1 \cos (q_1 + q_2) + \sin q_1 \sin (q_1 + q_2) \right) \\ &\quad + l_2^2 \left( \cos^2 (q_1 + q_2) + \sin^2 (q_1 + q_2) \right) \\ &= l_1^2 \cdot 1 + 2l_1 l_2 \cos (q_2) + l_2^2 \cdot 1 \\ &= l_1^2 + 2l_1 l_2 \cos q_2 + l_2^2 \end{aligned} \tag{5}$$

Based on Fig. 1 and Eq. 5, we can further derive

$$q_2 = \arccos \left( \frac{x_1^2 + x_2^2 - l_1^2 - l_2^2}{2l_1 l_2} \right) \tag{6}$$

$$\alpha = \arctan \frac{x_2}{x_1} \tag{7}$$

The Law of Cosines is a fundamental theorem that describes the relationship between the sides and angles of a triangle. By applying the Law of Cosines and referring to Fig. 1, we can further obtain

$$l_2^2 = l_1^2 + \left(x_1^2 + x_2^2\right) - 2l_1\sqrt{x_1^2 + x_2^2}\cos\beta, \tag{8}$$

$$\beta = \arccos\frac{x_1^2 + x_2^2 + l_1^2 - l_2^2}{2l_1\sqrt{x_1^2 + x_2^2}} \tag{9}$$

Then

$$q_1 = \begin{cases} \alpha - \beta, & q_2 > 0 \\ \alpha + \beta, & q_2 \leqslant 0 \end{cases} \tag{10}$$

Define the vectors $\boldsymbol{x} = \begin{bmatrix} x_1 & x_2 \end{bmatrix}$ and $\boldsymbol{q} = \begin{bmatrix} q_1 & q_2 \end{bmatrix}$. Then, the differential relationship $\mathrm{d}\boldsymbol{x} = \frac{\partial \boldsymbol{x}}{\partial g}\,\mathrm{d}\boldsymbol{q}$ can be expressed as $\mathrm{d}\boldsymbol{x} = \boldsymbol{J} \cdot \mathrm{d}\boldsymbol{q}$. Here, $\boldsymbol{J} = \frac{\partial \boldsymbol{x}}{\partial a}$ is defined as the Jacobian matrix

$$\boldsymbol{J} = \begin{bmatrix} \frac{\partial x_1}{\partial q_1} & \frac{\partial x_1}{\partial q_2} \\ \frac{\partial x_2}{\partial q_1} & \frac{\partial x_2}{\partial q_2} \end{bmatrix}$$

From Eq. (1), it can be seen that

$$\frac{\partial x_1}{\partial q_1} = -l_1\sin q_1 - l_2\sin\left(q_1 + q_2\right) \tag{11}$$

$$\frac{\partial x_1}{\partial q_2} = -l_2\sin\left(q_1 + q_2\right) \tag{12}$$

$$\frac{\partial x_2}{\partial q_1} = l_1\cos q_1 + l_2\cos\left(q_1 + q_2\right) \tag{13}$$

$$\frac{\partial x_2}{\partial q_2} = l_2\cos\left(q_1 + q_2\right) \tag{14}$$

Then

$$\boldsymbol{J(q)} = \begin{bmatrix} -l_1\sin\left(q_1\right) - l_2\sin\left(q_1 + \boldsymbol{q}_2\right) & -l_2\sin\left(q_1 + q_2\right) \\ l_1\cos\left(q_1\right) + l_2\cos\left(q_1 + q_2\right) & l_2\cos\left(q_1 + q_2\right) \end{bmatrix} \tag{15}$$

$$\boldsymbol{\dot{J}(q)} = \begin{bmatrix} -l_1\cos\left(q_1\right) - l_2\cos\left(q_1 + q_2\right) & -l_2\cos\left(q_1 + q_2\right) \\ -l_1\sin\left(q_1\right) - l_2\sin\left(q_1 + q_2\right) & -l_2\sin\left(q_1 + q_2\right) \end{bmatrix}\dot{q}_1$$

$$+ \begin{bmatrix} -l_2\cos\left(q_1 + q_2\right) & -l_2\cos\left(q_1 + q_2\right) \\ -l_2\sin\left(q_1 + q_2\right) & -l_2\sin\left(q_1 + q_2\right) \end{bmatrix}\dot{q}_2 \tag{16}$$

**Remark 2:** The Jacobian matrix $\boldsymbol{J}$, pivotal in robotics kinematics, serves as the derivative matrix that relates changes in joint angles $\boldsymbol{q}$ to changes in the end-effector's position $\boldsymbol{x}$ in Cartesian coordinates [54]. It encapsulates the entire set of first-order partial derivatives of the end-effector positions with respect to the joint angles. Essentially, the Jacobian matrix transforms joint velocity vectors into end-effector velocity vectors, enabling precise control and analysis of the manipulator's movement and dynamics in real-time applications.

## 2.2 Useful lemmas and properties

**Feature 1: symmetric and positive definite inertia matrix**

The inertia matrix $D_x(q)$ is symmetric and positive definite, meaning that the matrix is not only symmetric but also all its eigenvalues are positive. This attribute ensures the stability and controllability of the system because the positive definiteness indicates that the system will naturally tend toward a state of rest in the absence of external forces. This characteristic is crucial for the reliable operation of dynamic systems, as it implies that the system is inherently stable and will respond predictably to control inputs, making it easier to implement effective control strategies.

**Feature 2: Skew-symmetric property of the matrix $\dot{D}_x(q) - 2C_x(q, \dot{q})$**

The matrix $\dot{D}_x(q) - 2C_x(q, \dot{q})$ is skew-symmetric, indicating that the transpose of this matrix is its own negative. Skew-symmetry is crucial in dynamic models because it relates to energy conservation and the symmetry of system dynamics, helping to preserve the kinetic energy within the RM. This property ensures that any inherent angular momentum and motion energy are correctly accounted for without artificial gain or loss, thereby maintaining the physical fidelity of simulations and real world applications.

**Lemma 1 [55]:** For the function $V: [0, \infty) \to \mathbb{R}$, consider the differential inequality $\dot{V} \le -\alpha V + f$, holding for all $t \ge t_0 \ge 0$. The solution to this inequality is given by

$$V(t) \le e^{-\alpha(t-t_0)} V(t_0) + \int_{t_0}^{t} e^{-\alpha(t-\tau)} f(\tau) \mathrm{d}\tau \tag{17}$$

where $\alpha$ is any constant. This lemma states that the value of $V(t)$ can be bounded by the exponentially decayed value of $V(t_0)$, adjusted by $e^{-\alpha(t-t_0)}$, added to the integral from $t_0$ to $t$ of the function $f(\tau)$, each component of which is weighted by an exponentially decaying factor $e^{-\alpha(t-\tau)}$.

# 3 Robust sliding mode controller design

## 3.1 Control objective

The primary control objective of this study is to design and implement a robust SMC system for RMs, focusing on achieving precise trajectory tracking of the end effector under various operational conditions. By formulating a dynamic Cartesian coordinate model and incorporating customized SMC laws, the control strategy aims to address uncertainties and disturbances in complex engineering environments, ensuring reliable and efficient robotic performance. The proposed control approach is rigorously validated through extensive numerical simulations, which demonstrate its robustness in maintaining accurate trajectory tracking despite external perturbations and system dynamics.

## 3.2 A robot manipulator model

**Remark 3:** In RM modeling, it is essential to convert the conventional joint-based dynamics into Cartesian coordinates for effective end-effector control. Using the Jacobian matrix, joint variables and torques are transformed into dynamics that directly govern the end effector's position, as expressed by the equation $D_x(q)\ddot{x} + C_x(q, \dot{q})\dot{x} + G_x(q) = F_x$. This transformation simplifies control algorithms and improves the robot's interaction with its environment, focusing on stability, energy efficiency, and precision—key factors for high-performance operation in complex and dynamic workspaces.

**Remark 4:** To achieve precise control of the end effector's position, the traditional joint angle-based dynamics are transformed into end effector position-based equations using the

Jacobian matrix. This approach directly links the end effector's position to joint dynamics, simplifying control design and enhancing the robot's precision and efficiency in complex tasks.

Here, we consider a rigid robotic manipulator with $n$ joints, whose dynamic characteristics are as

$$D(q)\ddot{q} + C(q,\dot{q})\dot{q} + G(q) = \tau \tag{18}$$

where $q \in R^n$ represents the joint position variables, the vector $\tau \in R^n$ represents the joint torques applied by the actuators, $D(q) \in \mathbf{R}^{n\times n}$ is a symmetric and positive definite inertia matrix, ensuring the system's stability and responsiveness; $C(q,\dot{q}) \in \mathbf{R}^{n\times n}$ includes the Coriolis and centrifugal forces, which are crucial for precise control of the manipulator, $G(q) \in R^n$ represents the vector of gravitational forces impacting the RM, particularly vital in vertical or inclined operations.

In a static equilibrium state, a linear mapping relationship exists between the forces $F_x$ acting on the endeffector of a RM and the torques $\tau$ applied at the joints. This relationship can be elucidated through the application of the principle of virtual work, which asserts that for equilibrium to be maintained

$$F_x = J^{-\mathrm{T}}(q)\tau \tag{19}$$

For ease of expression and computation, the Jacobian matrix $J(q)$ is simply denoted as $J$, and its inverse $J^{-1}(q)$ as $J^{-1}$. This simplification is based on the pivotal role of the Jacobian matrix in the dynamics of RMs, specifically in linking the joint velocities $\dot{q}$ to the velocities of the end effector $\dot{x}$, then

$$\dot{q} = J^{-1}\dot{x} \tag{20}$$

$$\ddot{x} = \dot{J}\dot{q} + J\ddot{q} = \dot{J}J^{-1}\dot{x} + J\ddot{q} \tag{21}$$

From Eqs. (20) to (21), we have

$$\ddot{q} = J^{-1}\left(\ddot{x} - \dot{J}J^{-1}\dot{x}\right) \tag{22}$$

Substituting the above expression into Eq. (18) yields the following formula:

$$D(q)J^{-1}\left(\ddot{x} - \dot{J}J^{-1}\dot{x}\right) + C(q,\dot{q})J^{-1}\dot{x} + G(q) = \tau \tag{23}$$

Then

$$D(q)J^{-1}\ddot{x} - D(q)J^{-1}\dot{j}J^{-1}\dot{x} + C(q,\dot{q})J^{-1}\dot{x} + G(q) = \tau \tag{24}$$

It can be further sorted out

$$D(q)J^{-1}\ddot{x} + \left(C(q,\dot{q}) - D(q)J^{-1}\dot{J}\right)J^{-1}\dot{x} + G(q) = \tau \tag{25}$$

Sequentially,

$$J^{\mathrm{T}}\left(D(q)J^{-1}\ddot{x} + \left(C(q,\dot{q}) - D(q)J^{-1}\dot{j}\right)J^{-1}\dot{x} + G(q)\right) = J^{-\mathrm{T}}\tau \tag{26}$$

thus

$$D_x(q)\ddot{x} + C_x(q,\dot{q})\dot{x} + G_x(q) = F_x \tag{27}$$

where

$$D_x(q) = J^{-T}D(q)J^{-1} \tag{28}$$

$$C_x(q, \dot{q}) = J^{-T}\left(C(q, \dot{q}) - D(q)J^{-1}\dot{J}\right)J^{-1} \tag{29}$$

$$G_x(q) = J^{-T}G(q) \tag{30}$$

## 3.3 Design of sliding mode controller

**Remark 5:** This robust sliding mode controller for RMs is designed to ensure accurate trajectory tracking in the workspace. By introducing an error vector $e(t)$ and a corrective reference velocity $\dot{x}_r(t)$, the controller dynamically adjusts to reduce the discrepancy between the desired and actual trajectories. The control law integrates the robot's dynamics, modified by the Jacobian matrix, to counteract motion deviations effectively. A Lyapunov function guarantees system stability and exponential convergence, improving task performance and reliability.

Assume $x_d(t)$ is the ideal trajectory that the robotic manipulator should follow in the workspace, where $\dot{x}_d(t)$ and $\ddot{x}_d(t)$ represent the ideal velocity and acceleration, respectively. Define the error vector $e(t)$ as the difference between the ideal and actual trajectories, that is,

$$e(t) = x_d(t) - x(t) \tag{31}$$

Further, define the reference velocity $\dot{x}_r(t)$ as the ideal velocity plus a control input $\Lambda(t)$, and set the sliding mode variable $s(t)$ as the difference between the reference and actual velocities to assess tracking performance.

$$\dot{x}_r(t) = \dot{x}_d(t) + \Lambda(t) \tag{32}$$

$$s(t) = \dot{x}_r(t) - \dot{x}(t) = \dot{e}(t) + \Lambda(t) \tag{33}$$

The controller is designed as

$$F_x = D_x(q)\ddot{x}_r + C_x(q, \dot{q})\dot{x}_r + G_x(q) + Ks \tag{34}$$

where $K$ is a positive definite matrix, ensuring system stability and control accuracy.

Substituting the control law (34) into Eq. (27), we obtain the following equation:

$$D_x(q)\ddot{x} + C_x(q, \dot{q})\dot{x} + G_x(q) = D_x(q)\ddot{x}_r + C_x(q, \dot{q})\dot{x}_r + G_x(q) + Ks \tag{35}$$

Let

$$\dot{x} = \dot{x}_r - s \tag{36}$$

and take Eq. (36) into Eq. (35), we get

$$D_x(q)\dot{s} + C_x(q, \dot{q})s + Ks = 0 \tag{37}$$

Since $D_x(q)$ is symmetric and positive definite, a Lyapunov function can be defined to analyze the stability of the system.

$$V = \frac{1}{2}s^{\mathrm{T}}D_x(q)s \tag{38}$$

## 4 Simulation verification

In the simulation experiments of this study, the simulations were conducted on a machine with the following specifications: Intel i7-9700K processor (3.6 GHz, 8 cores), 16 GB DDR4 RAM, and NVIDIA GeForce RTX 2080 GPU, running Windows 10 Pro. The control algorithms were implemented in MATLAB 2023a with Simulink for system modeling and simulation. This configuration ensured efficient computation for the tasks involved. Through this simulation platform, the effectiveness and stability of the control algorithm can be further evaluated, thereby providing reliable data support and theoretical basis for subsequent practical applications.

Consider a planar two-joint RM, whose dynamics equation can be expressed as:

$$D(q)\ddot{q} + C(q,\dot{q})\dot{q} + G(q) = \tau \tag{39}$$

where

$$G(q) = \begin{bmatrix} m_1 g \cos q_1 + m_5 g \cos(q_1 + q_2) \\ m_5 g \cos(q_1 + q_2) \end{bmatrix} \tag{40}$$

$$C(q,\dot{q}) = \begin{bmatrix} -m_3 \dot{q}_2 \sin q_2 & -m_3(\dot{q}_1 + \dot{q}_2)\sin q_2 \\ m_3 \dot{q}_1 \sin q_2 & 0.0 \end{bmatrix} \tag{41}$$

$$D(q) = \begin{bmatrix} m_1 + m_2 + 2m_3 \cos q_2 & m_2 + m_3 \cos q_2 \\ m_2 + m_3 \cos q_2 & m_2 \end{bmatrix} \tag{42}$$

In the above equation, the values of $m_i$ are given by the following formula:

$$M = \begin{bmatrix} m_1 & m_2 & m_3 & m_4 & m_5 \end{bmatrix}^{\mathrm{T}}$$

$$P = \begin{bmatrix} p_1 & p_2 & p_3 & p_4 & p_5 \end{bmatrix}^{\mathrm{T}}$$

$$L = \begin{bmatrix} l_1^2 & l_2^2 & l_1 l_2 & l_1 & l_2 \end{bmatrix}^{\mathrm{T}}$$

Here, $p_1$ represents the load, while $l_1$ and $l_2$ denote the lengths of joint 1 and joint 2 , respectively. The vector $P$ is defined as P $= \begin{bmatrix} 1.46 & 0.402 & 0.75 & 3.025 & 1.095 \end{bmatrix}$, with the lengths $l_1 = 1$ and $l_2 = 1$.

In the Cartesian space, the desired tracking trajectory is defined as $x_{d1} = \cos t$ and $x_{d2} = \sin t$. This trajectory describes a circle with a radius of 1.0 , centered at the coordinates $(0.0, 0.0)$m. The initial conditions are set as $x(0) = \begin{bmatrix} 1.0 & 1.0 \end{bmatrix}$ and $\dot{x}(0) = \begin{bmatrix} 0.0 & 0.0 \end{bmatrix}$.

Since the tracking trajectory is specified in Cartesian coordinates, rather than in joint space angles, it is necessary to convert the end effector coordinates $(x_1, x_2)$ into joint angles $(q_1, q_2)$ using Eqs. (6) and (10). This conversion is crucial for ensuring accurate control and trajectory tracking in RM systems. The formula for the sliding mode controller is given by Eq. (34), with the controller gains selected as

$$K = \begin{bmatrix} 20 & 0 \\ 0 & 20 \end{bmatrix} \quad \text{and} \quad \Lambda = \begin{bmatrix} 10 & 0 \\ 0 & 10 \end{bmatrix}.$$

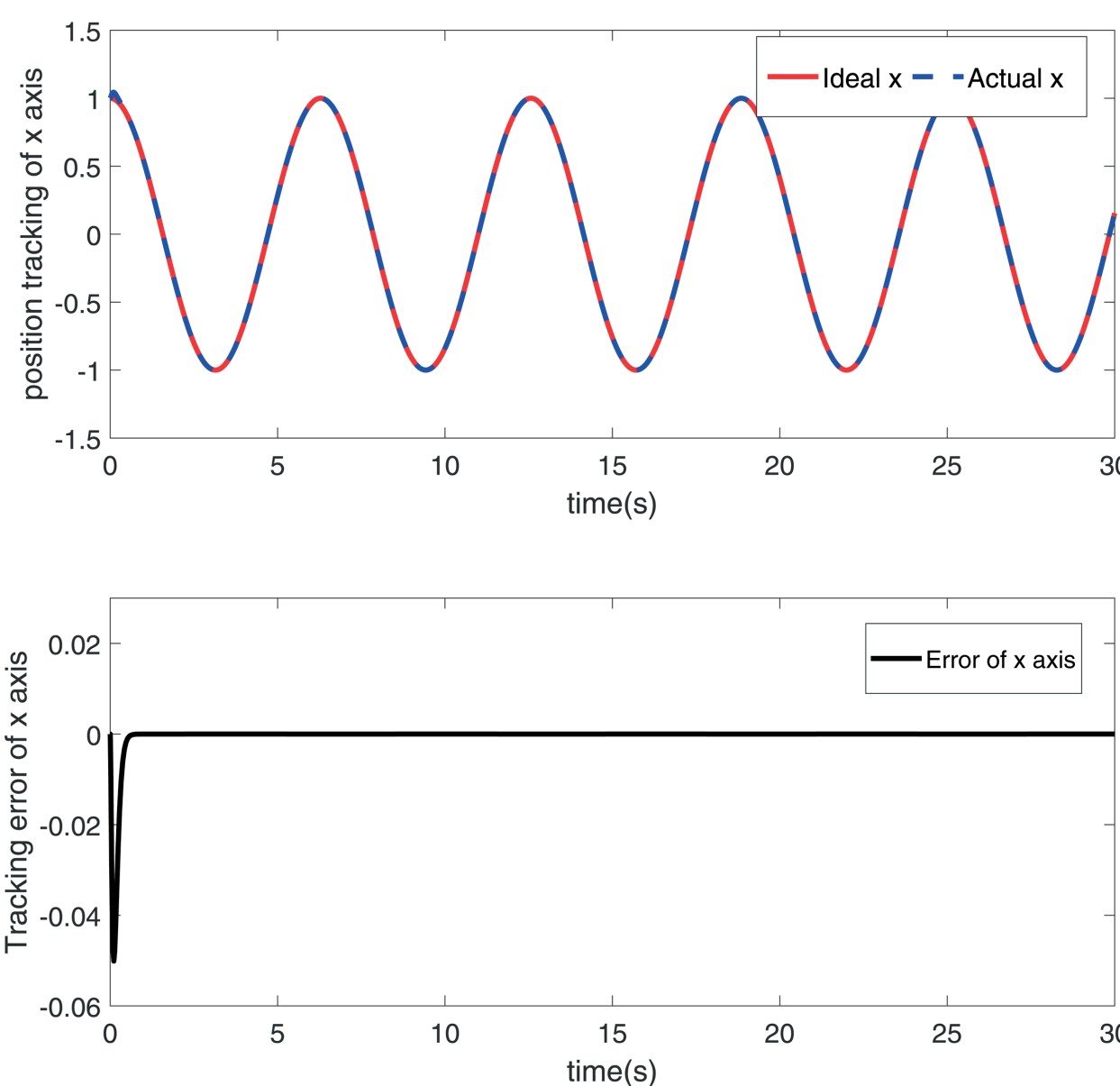

**Fig 2. The position tracking and error diagram of the *x*-coordinate of the end joint node of the RM.**

The simulation results are shown in Figs. 2–8.

Figs. 2 and 3 depict the position tracking and errors of a robotic manipulator's (RM) end joint along the x-axis and y-axis, respectively. In Fig. 2, the red line (Ideal x) represents the ideal position along the x-axis, while the blue dashed line (Actual x) shows the measured actual position. The proximity of these two lines indicates that the actual position closely follows the target trajectory, with the black solid line illustrating the position error. As seen in Fig. 2, the error fluctuates around zero, ranging within approximately ±0.02, demonstrating the system's high precision and stability on the x-axis. Fig. 3 shows similar tracking on the y-axis, where the red line (Ideal y) represents the ideal position, and the blue dashed line (Actual y) shows the actual position. The position error on the y-axis, depicted by the black

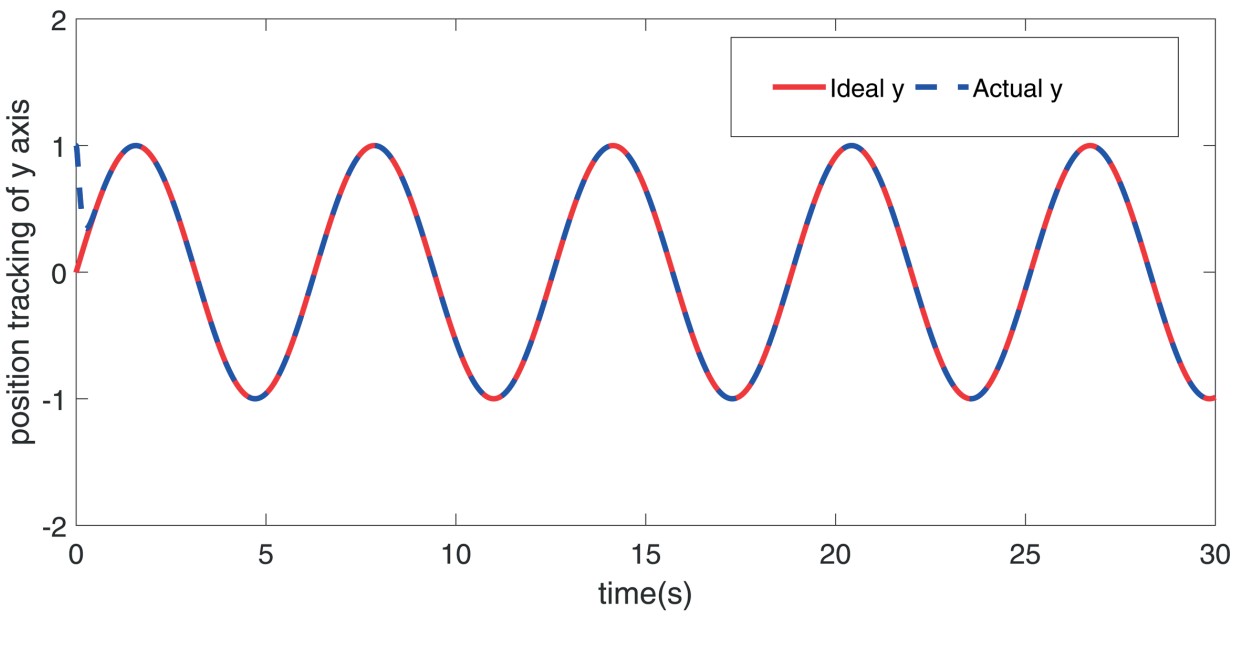

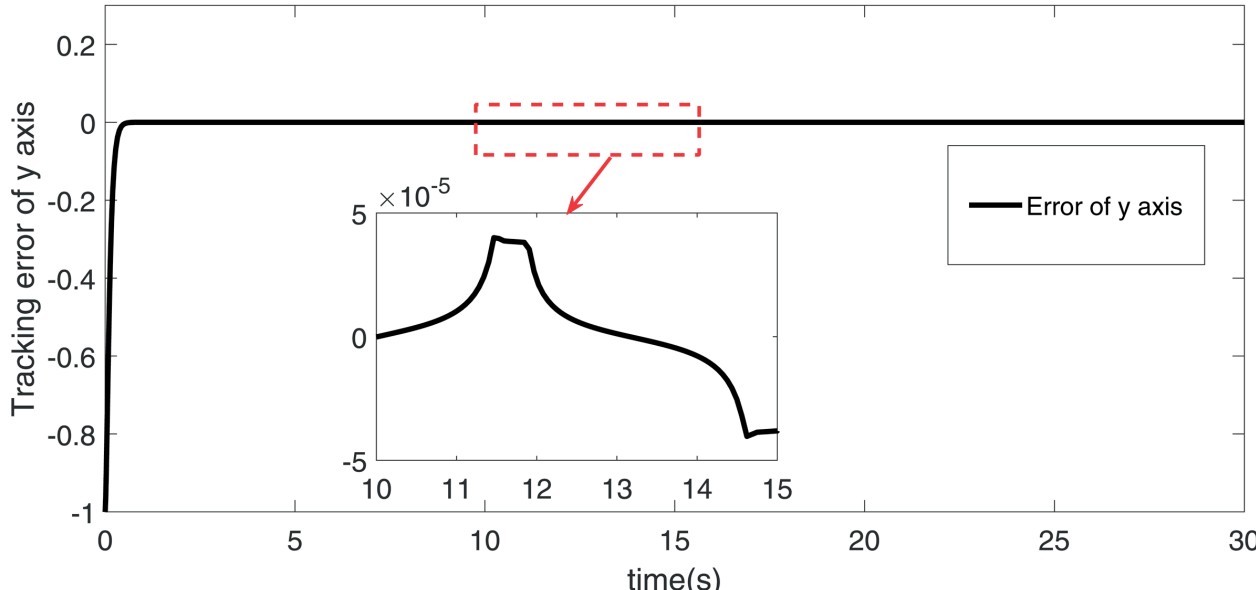

**Fig 3. The position tracking and error diagram of the *y*-coordinate of the end joint node of the RM.**

solid line, shows minor fluctuations, particularly between 10 and 15 seconds, but remains within a small range. The results in Figs. 2 and 3 confirm that the robust sliding mode control (SMC) strategy successfully achieves precise trajectory control of the RM's end effector along both axes. Fig. 4 and 5 further demonstrate the velocity tracking and error conditions of the RM's end joint along the x-axis and y-axis, respectively. In Fig. 4, the red line (Ideal

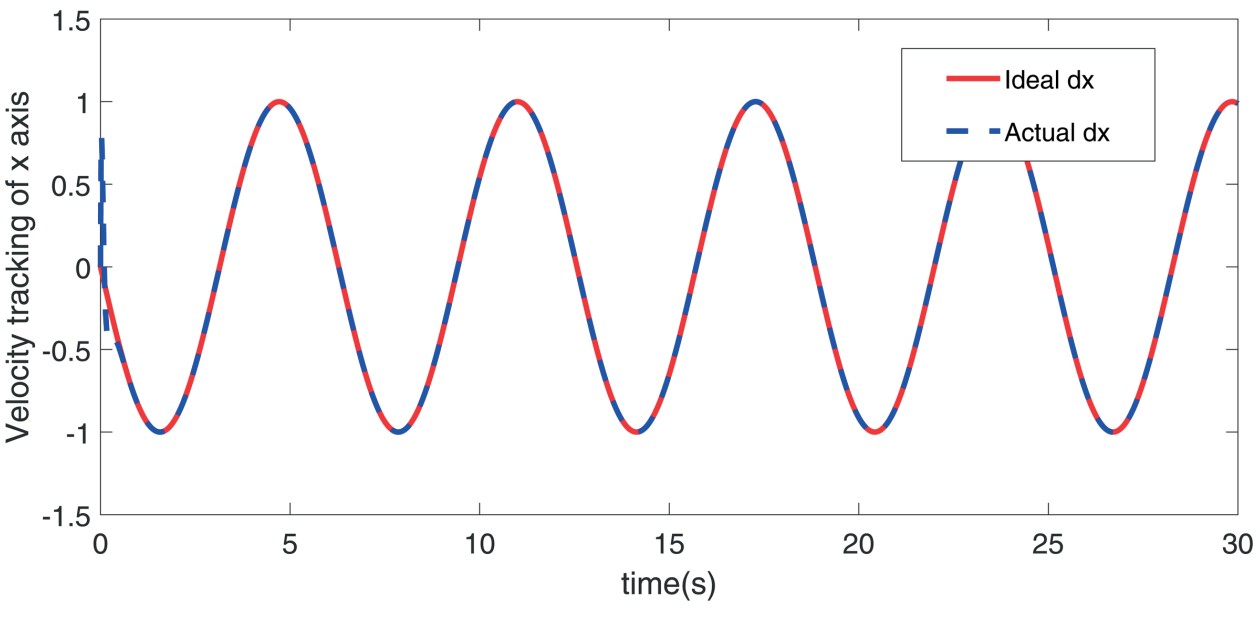

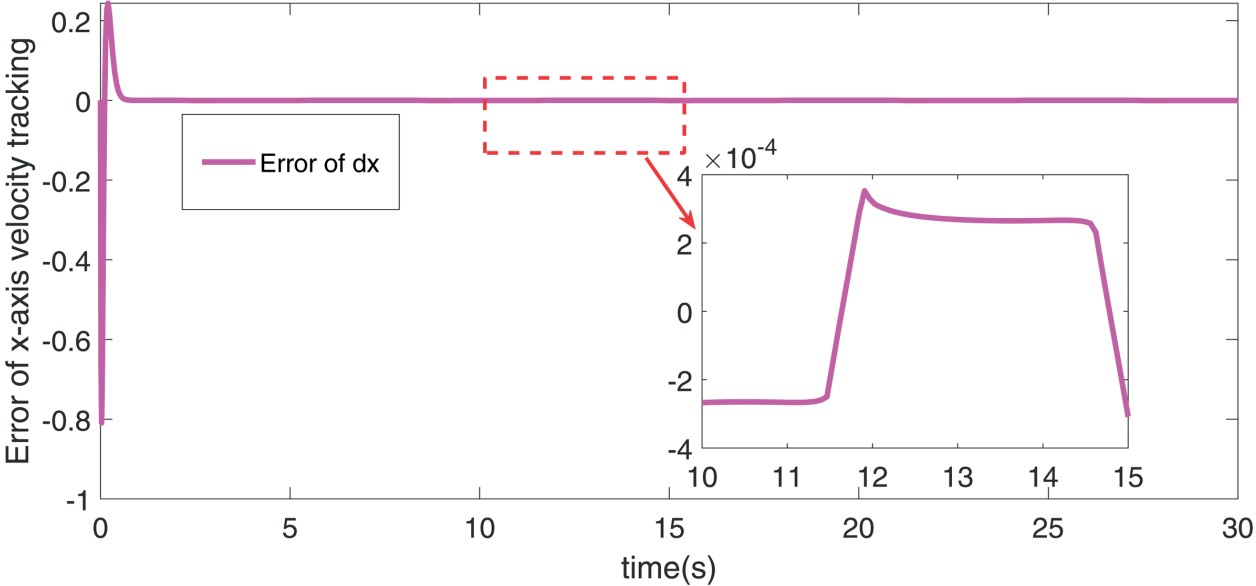

**Fig 4. The velocity tracking and error diagram of *x*-coordinate of the end joint node of RM.**

dx) shows the ideal velocity along the x-axis, while the blue dashed line (Actual dx) represents the measured velocity. The close alignment of the actual velocity with the ideal velocity indicates excellent velocity control. The error curve, mostly within ±0.2, stays near zero from 10 to 15 seconds, highlighting the system's high accuracy in controlling velocity along the x-axis. In Fig. 5, the red line (Ideal dy) represents the ideal velocity along the y-axis, with the blue dashed line (Actual dy) showing the actual velocity. Although there are minor fluctuations in the actual velocity, it generally follows the ideal trajectory. The black solid line

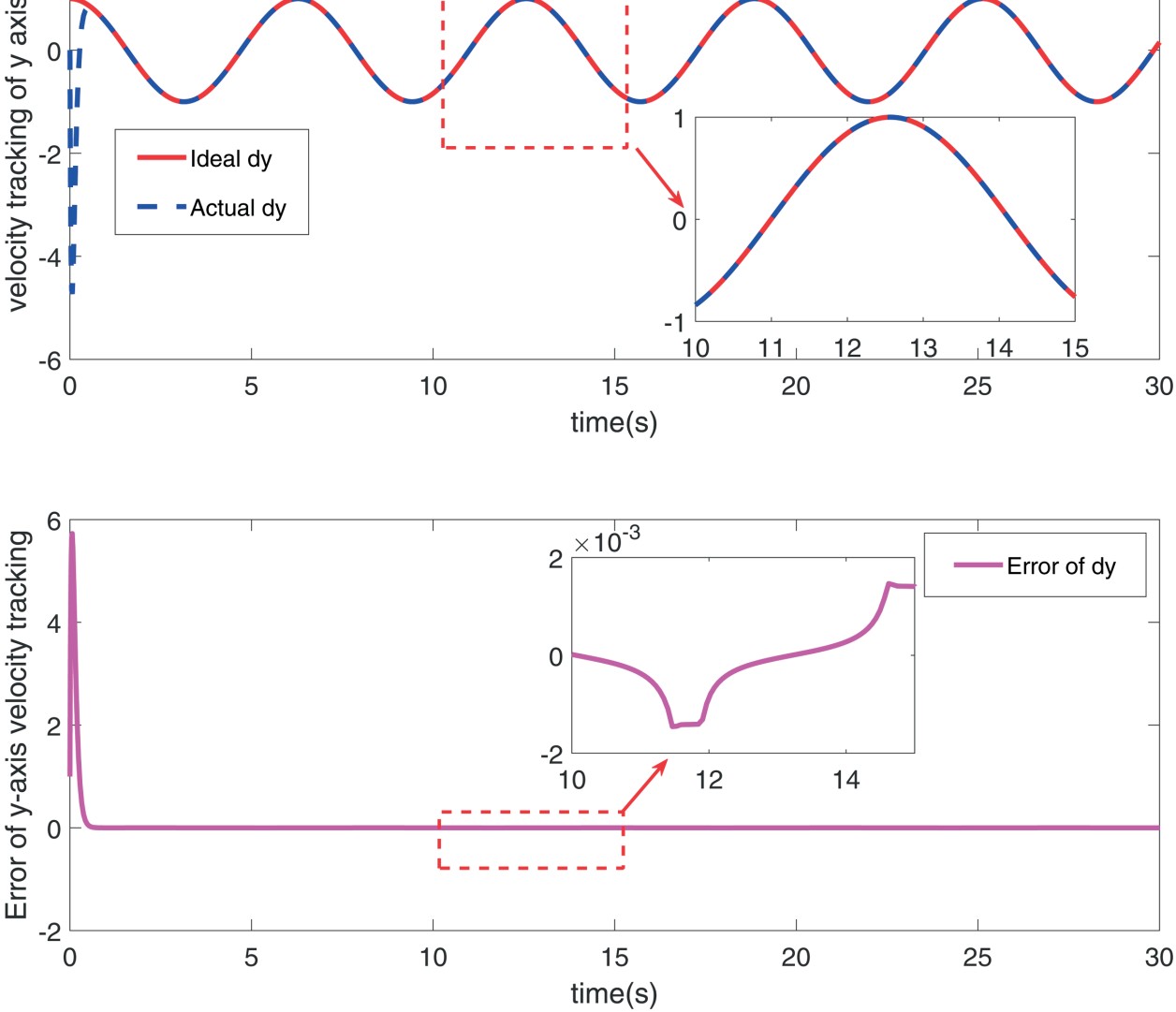

**Fig 5. The velocity tracking and error diagram of *y*-coordinate of the end joint node of RM.**

shows the velocity error, which remains stable, though with some fluctuations between 10 and 15 seconds. This suggests potential areas for optimization, especially for more dynamic conditions. Overall, Figs. 4 and 5 demonstrate that the proposed control strategy effectively ensures precise velocity tracking along both axes, with brief periods of increased error on the y-axis. These results highlight the system's strong performance in dynamic conditions, with particularly high accuracy and responsiveness in velocity control.

Figs. 6 and 7 respectively display the response diagrams of control input force $F_x$ and control input torque $\tau$ for two links of a robot. Fig. 6 illustrates the control input forces $F_x$ applied to the two links. The red solid line represents the control input force for the first link, while

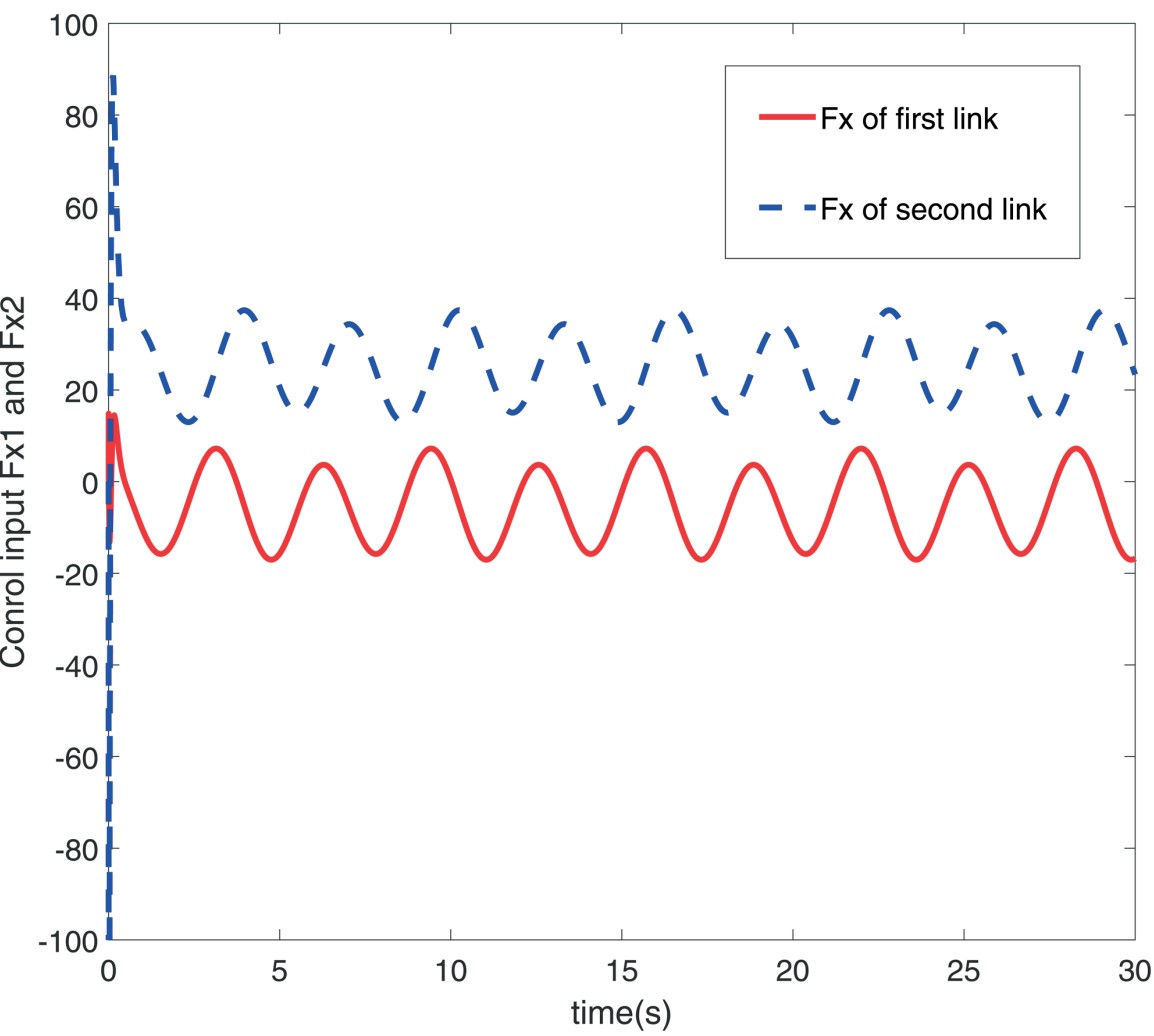

**Fig 6. Response diagram of control input $F_x$.**

the blue dashed line represents the force applied to the second link. It is evident that the control input forces for the two links display distinct fluctuation patterns, with the first link experiencing larger amplitude variations compared to the smaller fluctuations observed for the second link. Fig. 7 illustrates the control input torques for the two links. The red solid line represents the control input torque for the first link, and the blue dashed line for the second link. The torque curves for the two links are nearly inverse of each other, indicating that torque adjustments are made to maintain balance and stability of the mechanical system during operations. Figs. 6 and 7 present the dynamic responses of the control input forces and torques for the two robot links. These figures highlight the interaction between the links and their reaction to external control commands during precise operations. The control inputs exhibit distinct fluctuation patterns and inverse adjustments, showcasing the system's ability to maintain dynamic equilibrium and execute accurate tasks. Fig. 8 illustrates the trajectory tracking performance of the robot's end effector along the x and y axes in a 2D space. The red solid line (Ideal trajectory) represents the desired path, while the blue dashed line (Actual trajectory) shows the real trajectory followed. As seen in Fig. 8, the ideal trajectory forms a

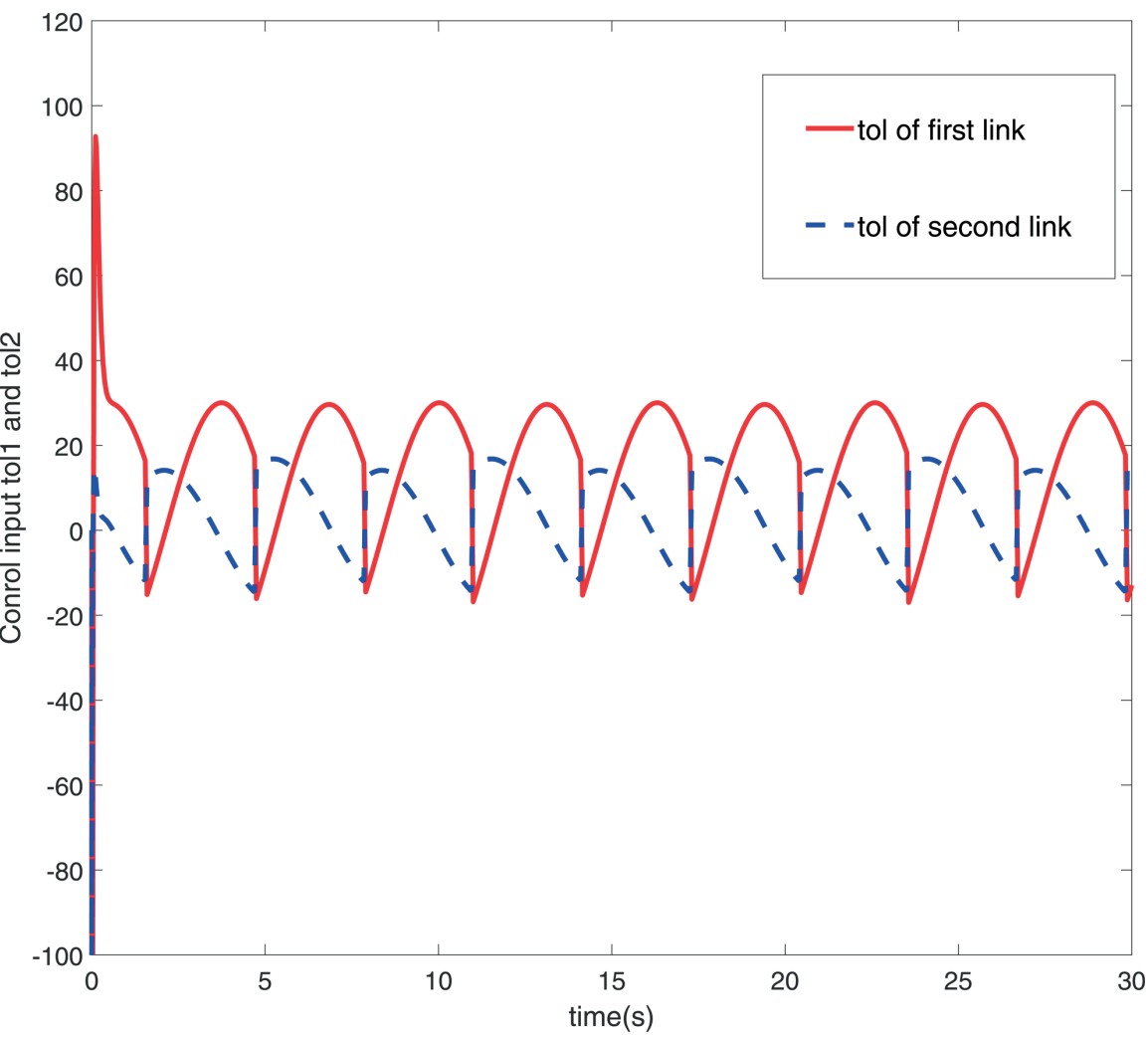

**Fig 7. Response diagram of control input $\tau$.**

perfect circle, and the actual trajectory closely matches the ideal path in most regions. However, there is a noticeable deviation on the right side of the figure, forming a distinct arc. This suggests that the control system may experience response delay or lack execution accuracy in that part of the trajectory. Additionally, the small box in the lower left corner provides a magnified view of the trajectory, further highlighting the deviation of the actual trajectory from the ideal trajectory in this specific area. Overall, the actual trajectory is very close to the ideal trajectory, indicating a high tracking accuracy of the control system.

## 5 Discussion and future work

This study presents a detailed exploration of robotic manipulators (RMs) for automation tasks in hazardous environments, proposing a dynamic Cartesian coordinate model to control the end-effector trajectory with high precision. By developing robust SMC laws, we successfully mapped the control strategy to joint torques, ensuring that the end-effector follows the pre-defined path accurately. The proposed approach enhances production efficiency and product

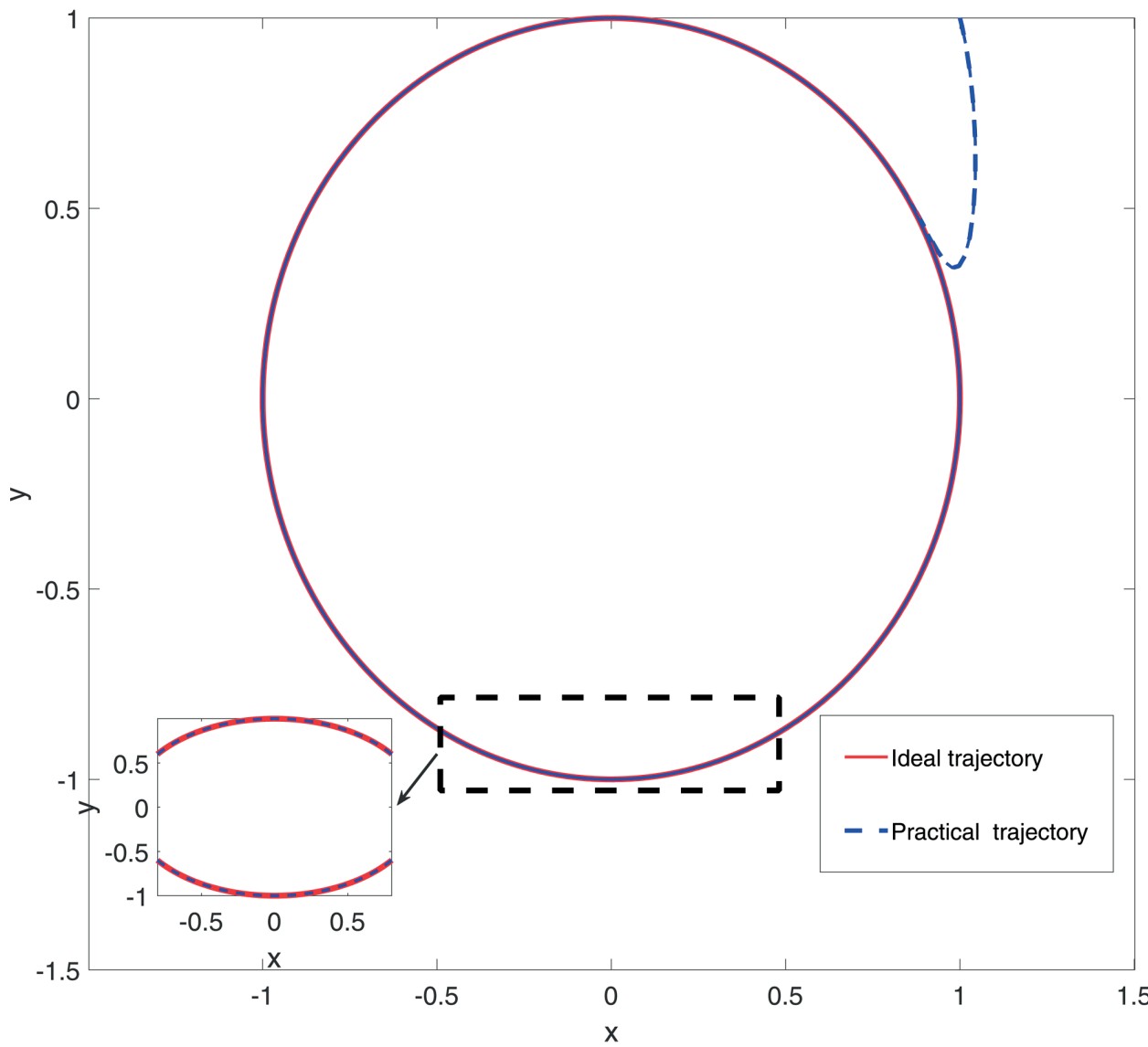

**Fig 8. The display diagram of *x,y* trajectory tracking effect.**

quality while contributing valuable insights into the spatial dynamics of robotic systems in practical engineering contexts.

However, the current study is limited to simulation-based validation. To further substantiate the findings, future work will focus on experimental validation using a physical robotic manipulator. Challenges in transitioning from simulation to physical implementation include addressing actuator and sensor inaccuracies, dealing with real-time computational constraints, and mitigating external disturbances. To this end, we plan to conduct real-world experiments to evaluate the robustness of the proposed SMC strategy in dynamic environments and identify any practical issues that may arise in physical implementations. We anticipate that such experiments will provide more reliable data on the control system's performance and help refine the control laws for real-world applications.

Additionally, we will explore how our approach can be integrated with other advanced control techniques to further enhance robustness against model uncertainties and disturbances. Moreover, future research will investigate the potential for collaborative multi-robot systems, such as coordinating robotic manipulators with autonomous drones, to improve the efficiency and flexibility of automated production and service models in real-world industrial settings.

## Supporting information

**S1. Paper program**.
(PDF)

## Author contributions

**Data curation:** Beibei Su.

**Methodology:** Beibei Su.

**Writing – original draft:** Beibei Su.

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
