## [Decision Letter · Decision Letter 0]

20 Dec 2024

PONE-D-24-49978Tracking control of robotic manipulator end-effector trajectory based on robust sliding mode methodPLOS ONE

Dear Dr. Su,

Thank you for submitting your manuscript to PLOS ONE. After careful consideration, we feel that it has merit but does not fully meet PLOS ONE’s publication criteria as it currently stands. Therefore, we invite you to submit a revised version of the manuscript that addresses the points raised during the review process.

We look forward to receiving your revised manuscript.

Kind regards,

Jamshed Iqbal, PhD

Academic Editor

PLOS ONE

“This research is funded by the General Project of Philosophy and Social Sciences Research in Jiangsu Province Higher Education Institutions (Project Title: "Research on Entrepreneurial Models for College Students in Higher Vocational Colleges Based on 'Co-Creation between Teachers and Students," Project No. 2024SJYB0726).”

4. We are unable to open your Supporting Information file [S1.zip]. Please kindly revise as necessary and re-upload.

Additional Editor Comments:

- Include a more thorough analysis of error such as by including IAE, ITAE and ISE, etc.

- Figure 1 can be labeled to convey more useful information.

- SMC and several of its variants suffer from chattering. Discuss this phenomenon with reference to 10.1371/journal.pone.0260480

- Discussion on applications of RMs in industrial applications mentioned in Line 5 could benefit from the reference such as 'An autonomous image-guided robotic system simulating industrial applications'

- Instead of referring to the color of waveforms in the Results section, please refer to the lines w.r.t. their shape (solid line, dotted line, etc.)

- Update the literature review by including notable works on RM control such as 'Adaptive Backstepping Based Sensor and Actuator Fault Tolerant Control of a Manipulator'.

- Results need more rigorous and critical discussion.

- Please thoroughly proofread the paper for typos and linguistic improvements e.g. 1-4 on Line should be [1-4].

Reviewers' comments:

Reviewer's Responses to Questions

**Comments to the Author**

1. Is the manuscript technically sound, and do the data support the conclusions?

Reviewer #1: Yes

Reviewer #2: Yes

Reviewer #3: Yes

2. Has the statistical analysis been performed appropriately and rigorously? 

Reviewer #1: No

Reviewer #2: Yes

Reviewer #3: Yes

3. Have the authors made all data underlying the findings in their manuscript fully available?

Reviewer #1: Yes

Reviewer #2: Yes

Reviewer #3: Yes

4. Is the manuscript presented in an intelligible fashion and written in standard English?

Reviewer #1: Yes

Reviewer #2: Yes

Reviewer #3: Yes

5. Review Comments to the Author

Reviewer #1: The contribution of the paper can be summarized as follows:

This article presents a sliding mode control strategy for controlling the robotic manipulator end-effector in Cartesian space.

Furthermore, there are serious major comments together with some minor ones that need to be addressed before the paper can reach the state to of acceptance.

The major comments are as follows:

1. The cartesian space model for robotic manipulator have been addressed and published previously in [P. Sanchez-Sanchez, 2005] DOI: 10.1109/IROS.2005.1545518

2. Please, insert the block diagram that explains the experimental simulations.

3. Please, insert a comparative study to show the advantages of the proposed scheme.

Where the sliding mode control have been developed for the robotic manipulator in different ways. see [Seung-Hun Han,2021] https://doi.org/10.3390/app11093919

4. The proposed controller should be tested under system uncertainty and disturbance to show the robustness.

Reviewer #2: Please correct any grammatical errors or typos and ensure consistent use of notations and abbreviations throughout the text. Additionally, provide a more detailed and thorough explanation to help readers fully understand the model and its operations."

--T represents the torque as stated in Equation 18, whereas the control effort in Equation 38 is denoted as . Could you clarify how the control effort, in terms of torque, is applied to the actuators?

--Could you provide the stability analysis and include further commentary to enhance understanding of the system's stability?

--As stated in the abstract, by mapping the control laws to the joint torques, we calculate the actual torque required. Could you also highlight the performance parameter values, such as the exact torque required, efficiency, and other relevant metrics?

--There are also other nonlinear controls like SMC that have not been cited in the paper that offer excellent performance in the field of control like

DOI: 10.1371/journal.pone.0256491

DOI: 10.1109/ACCESS.2021.3139041

--Please include a conclusion section before the discussion and future work sections to enhance readability for the readers.

Reviewer #3: Control techniques inherently suffer from uncertain disturbances. Referencing to 'Dynamic modeling and stabilization of surveillance quadcopter in space based on neuro fuzzy integral super twisting sliding mode control strategy', discuss in detail about possible mitigation strategies. Also, include results on how much the proposed control law could manage to reduce these disturbances?”

The comparative analysis is weak. A deeper comparison with other state-of-the-art methods, such as Adaptive FIT-SMC Approach for an Anthropomorphic Manipulator with Robust Exact Differentiator and Neural Network Based Friction Compensation, would give readers a clearer sense of how the proposed method stands in the broader context of robust control research.

The manuscript lacks a detailed discussion on how the various parameters of the proposed controller are selected and tuned. This is highly relevant. So, a more structured approach to tuning these parameters and explaining the trade-offs must be included.

The method's stability is validated using the Lyapunov theory, although this section could benefit from a more thorough discussion. Specifically, the paper could elaborate on the practical implications of the stability guarantees and whether they hold under real-world conditions, particularly under unmodeled dynamics or significant disturbances.

Discussion on sliding mode control in introduction could benefit from a recently reported reference e.g. doi.org/10.1080/00207179.2022. 2056514, doi.org/10.1016/j.isatra.2021.02.045, dx.doi.org/10.1007/s12555-019-0302-3.

Mathematical formalism initiated good, but soon, the manuscript gets messy and like from a different paper.

Detail the computer you used to perform the experiments.

Share an open-access repository to check the results. It will have more impact on your research. You can use GitHub, OceanCode, etc.

- Thoroughly proofread the paper for typos and linguistic improvements.

- Make sure that all abbreviations are elaborated/defined.

- Figures are needed to regenerate.

6. PLOS authors have the option to publish the peer review history of their article (what does this mean?). If published, this will include your full peer review and any attached files.

Reviewer #1: **Yes: **Dr. Nasr Elkhateeb

Reviewer #2: No

Reviewer #3: No

---

## [Author Response · Author response to Decision Letter 1]

27 Dec 2024

Please see the attachment uploaded.

---

## [Decision Letter · Decision Letter 1]

11 Feb 2025

PONE-D-24-49978R1Tracking control of robotic manipulator end-effector trajectory based on robust sliding mode methodPLOS ONE

Dear Dr. Su,

Thank you for submitting your manuscript to PLOS ONE. After careful consideration, we feel that it has merit but does not fully meet PLOS ONE’s publication criteria as it currently stands. Therefore, we invite you to submit a revised version of the manuscript that addresses the points raised during the review process. Please submit your revised manuscript by Mar 28 2025 11:59PM. If you will need more time than this to complete your revisions, please reply to this message or contact the journal office at plosone@plos.org. Please include the following items when submitting your revised manuscript:

We look forward to receiving your revised manuscript.

Kind regards,

Jamshed Iqbal, PhD

Academic Editor

PLOS ONE

Journal Requirements:

Additional Editor Comments:

The revised version of the paper has been improved. There are a few minor comments that need to be addressed before the paper can be considered for publication:

- Abstract needs to be rewritten. Please reduce the background theoretical details in the abstract and elaborate more on the methodology such as the details on SMC control law developed in the work.Also, summarise the control performance achieved quantitatively.

- Work is limited to the simulation environment. Include experimental results to support your findings. If this is not possible, at least include a critical discussion on how the work can be realised on a real physical robotic manipulator.

- The discussion on applications of robotic manipulators in the Introduction could benefit from literature such as 'An autonomous image-guided robotic system simulating industrial applications'

- Please label figure 1 so that it conveys more useful and clear information e.g. with words such as End-effector, Link(s), Joint(s), etc.

- Include a rigorous analysis of the error by taking parameters like IAE, ISE, ITAE etc.

- An interesting literature review is presented though there is a scope of including recently reported works. See 'Adaptive Backstepping Integral Sliding Mode Control of a MIMO Separately Excited DC Motor'.

- Explicitly state the assumptions made in the study.

- While reporting the literature review, write only the name of the first author followed by the et al. (instead of writing the names of all the authors).

- Please thoroughly proofread the paper for typos and other linguistic improvements.

Reviewers' comments:

Reviewer's Responses to Questions

**Comments to the Author**

1. If the authors have adequately addressed your comments raised in a previous round of review and you feel that this manuscript is now acceptable for publication, you may indicate that here to bypass the “Comments to the Author” section, enter your conflict of interest statement in the “Confidential to Editor” section, and submit your "Accept" recommendation.

Reviewer #1: All comments have been addressed

Reviewer #3: All comments have been addressed

2. Is the manuscript technically sound, and do the data support the conclusions?

Reviewer #1: Partly

Reviewer #3: Yes

3. Has the statistical analysis been performed appropriately and rigorously? 

Reviewer #1: Yes

Reviewer #3: Yes

4. Have the authors made all data underlying the findings in their manuscript fully available?

Reviewer #1: Yes

Reviewer #3: Yes

5. Is the manuscript presented in an intelligible fashion and written in standard English?

Reviewer #1: Yes

Reviewer #3: Yes

6. Review Comments to the Author

Reviewer #1: (No Response)

Reviewer #3: All my concens are answered. I have no more questions. It is recommended for publication in the revised form.

7. PLOS authors have the option to publish the peer review history of their article (what does this mean?). If published, this will include your full peer review and any attached files.

Reviewer #1: No

Reviewer #3: **Yes: **Safeer Ullah

---

## [Editor Report · Decision Letter 2]

14 Feb 2025

Tracking control of robotic manipulator end-effector trajectory based on robust sliding mode method

PONE-D-24-49978R2

Dear Dr. Su,

We’re pleased to inform you that your manuscript has been judged scientifically suitable for publication and will be formally accepted for publication once it meets all outstanding technical requirements.

Kind regards,

Jamshed Iqbal, PhD

Academic Editor

PLOS ONE

Additional Editor Comments (optional):

Th author has addressed all the suggested comments. The paper can be accepted.
---

## [Editor Report · Acceptance letter]

PONE-D-24-49978R2

PLOS ONE

Dear Dr. Su,

I'm pleased to inform you that your manuscript has been deemed suitable for publication in PLOS ONE. Congratulations! Your manuscript is now being handed over to our production team.

Kind regards,

on behalf of

Dr. Jamshed Iqbal

Academic Editor

PLOS ONE